# Comparing Blind and Ultrasound-Guided Retrobulbar Nerve Blocks in Equine Cadavers: The Training Effect

**DOI:** 10.3390/ani12020154

**Published:** 2022-01-09

**Authors:** Hanneke Hermans, Ralph A. Lloyd-Edwards, Aukje J. H. Ferrão-van Sommeren, Anne A. Tersmette, Jacobine C. M. Schouten, Filipe M. Serra Bragança, Johannes P. A. M. van Loon

**Affiliations:** 1Department of Clinical Sciences, Equine Sciences, Faculty of Veterinary Medicine, Utrecht University, Yalelaan 112-114, 3584 CM Utrecht, The Netherlands; anne.tersmette@gmail.com (A.A.T.); f.m.serrabraganca@uu.nl (F.M.S.B.); j.p.a.m.vanloon@uu.nl (J.P.A.M.v.L.); 2Department of Clinical Sciences, Diagnostic Imaging, Faculty of Veterinary Medicine, Utrecht University, Yalelaan 108, 3584 CM Utrecht, The Netherlands; R.A.Edwards@uu.nl; 3VetCT, St John’s Innovation Centre, Cowley Road, Cambridge CB4 0WS, UK; 4Diergeneeskundig Centrum Maas en Waal, Klepperheide 16, 6651 KM Druten, The Netherlands; aukjevansommeren@gmail.com; 5Department of Clinical Sciences, Anatomy and Physiology, Faculty of Veterinary Medicine, Utrecht University, Yalelaan 1, 3584 CM Utrecht, The Netherlands; j.schouten@uu.nl; 6Sporthorse Medical Diagnostic Centre (SMDC), Hooge Wijststraat 7, 5384 RC Bernheze, The Netherlands

**Keywords:** local, anesthetic, horse head, welfare, ophthalmic, pain management

## Abstract

**Simple Summary:**

Standing ophthalmic surgeries have become more and more common in horses. For these standing surgeries, the blind retrobulbar block is often used for anesthesia and akinesia of the eye. However, placing a retrobulbar block using this blind technique can lead to complications, for example, penetration of the globe, nerve injury or chemosis due to spreading of the local anesthetic in the region of the globe. For this reason, it might be better to perform the retrobulbar block using guidance by ultrasound. Ultrasound-guided retrobulbar block has only been described twice in the literature using equine cadavers. Comparison of the ultrasound-guided peribulbar technique to the blind technique has only been carried out once. Furthermore, the learning curve of ultrasound-guided retrobulbar nerve block placement has not been evaluated. Our study aimed to compare the blind and ultrasound-guided approaches to retrobulbar block placement in horses and to evaluate the success and complication rates, analyzing the effect of training on ultrasound guidance. A trend towards a significant improvement in accuracy was seen for ultrasound guidance, and larger scale follow-up studies might show a significant training effect on the use of ultrasound in retrobulbar nerve block placement and that the use of ultrasound guidance could be promising.

**Abstract:**

In standing ophthalmic surgery in horses, a retrobulbar nerve block (RNB) is often placed blindly for anesthesia and akinesia. The ultrasound (US)-guided RNB may have fewer complications, but the two techniques have only been compared once in equine cadavers. This study compares the techniques for success and complication rates and analyzes the effect of training on US guidance. Twenty-two equine cadavers were divided into three groups: blind RNBs were performed bilaterally in eight cadavers, US-guided RNBs were performed bilaterally in seven cadavers, and after US-guided training, blind RNBs were performed bilaterally in seven cadavers. All RNBs were performed by the same two inexperienced operators, and a combination of contrast medium (CM; 1.25 mL) and methylene blue dye (1.25 mL) were injected (2.5 mL total volume). Needle positioning in the periorbita and the distance of the CM to the optic foramen were assessed using computerized tomography (CT). Dye spreading was evaluated by dissection. In group 1, 37.5% of the injections were in the optimal central position in the periorbita; in group 2, 75% and in group 3, 71.4%. There was no significant difference between the groups regarding needle position (groups 1 and 2 *p* = 0.056; groups 1 and 3 *p* = 0.069, groups 2 and 3 *p* = 0.8). The mean CM distribution distance was not significantly different between all groups. Group 1 had 18.75% intraocular injections versus 0% in group 2 and 7.1% in group 3 (not significant). US guidance showed no significant increases in accuracy nor decreases in complications. However, the effects on accuracy showed a trend towards significant improvement, and larger scale follow-up studies might show significant training effects on US guidance.

## 1. Introduction

Standing ophthalmic procedures, increasingly popular in everyday equine practice, require adequate local anesthesia and analgesia. For most standing ophthalmic procedures, such as enucleation or corneal surgeries, a retrobulbar nerve block (RNB) is performed blindly for anesthesia and akinesia. In human medicine, blind placement of a retrobulbar block, historically the gold standard, has been mostly replaced by ultrasound (US)-guided techniques, which have fewer severe complications [1,2,3,4,5]. In human medicine, compared to blind techniques in other locoregional procedures, US-guided techniques have demonstrated reduced complication rates and improved block quality [6,7]. Moreover, they have improved the learning curve of clinicians [8].

For RNB placement in horses, several techniques have been described, including the blind and US-guided technique [9,10,11]. This US-guided technique has recently been described for various procedures in horses but only twice for US-guided RNB using equine cadavers [12,13,14,15,16]. Comparison of the US-guided peribulbar technique to the blind technique in horses has only been performed once [16]. Furthermore, the learning curve of US-guided RNB placement has not been evaluated. The aim of our study was, thus, to compare the blind technique with the US-guided technique in horses and to determine whether ultrasound guidance is significantly useful in RNB placement. We expect that ultrasound guidance will improve the learning curve of inexperienced operators.

## 2. Materials and Methods

### 2.1. Animals

Twenty-two clinically healthy Shetland ponies, comprising 21 mares and 1 gelding, with a mean age of 4.8 years (range 4–10.5 years) and a mean body weight of 167.4 kg (range 138–204 kg), without signs of ophthalmic disease, were euthanized for reasons other than this study and, afterwards, heads were removed from the trunk.

Of the 22 equine cadavers both eyes were used, so, in total, 44 retrobulbar injections were performed (retrobulbar injections will be referred to in the rest of the manuscript as retrobulbar nerve blocks (RNBs)). All cadaver heads were used fresh or frozen and then thoroughly thawed before injection. Hair was removed bilaterally around the area of the supraorbital fossa and the area cleaned with alcohol.

All RNBs were divided equally between an equine clinician (A.J.H.F.-v.S.) and a clinical student of veterinary medicine (A.A.T). Neither individual had prior experience performing the RNB. Each performed the RNB on one eye of each head, randomly choosing the left or right side.

All RNBs were supervised by a board-certified veterinary anesthesiologist (J.P.A.M.v.L). The injections were performed with a combination of contrast medium (CM; Iobitridol, 768 mg/mL Xenetix^®^, Guerbet, Gorinchem, The Netherlands; 1.25 mL) and methylene blue dye (1.25 mL) using a 20 G 75 mm long spinal needle (B. Braun Melsungen AG., Melsungen, Germany). The total injection volume was 2.5 mL.

The cadaver heads were divided into three groups: around six weeks passed between injections in group 1 and 2 and around three months between group 2 and 3. Time between administration of the RNB and cadaver evaluation was standardized for all cadaver heads. Immediately after placement of the RNB, CT was performed, followed by dissection of the cadaver heads. Time between RNB and dissection was two hours maximum.

#### 2.1.1. Group 1—Blind Technique

RNBs were performed using the blind technique in eight cadaver heads (16 cadaver orbits). The spinal needle was placed using the dorsal (supraorbital) approach and placed caudal to the center of the dorsal orbital rim [17]. It was advanced into the retrobulbar space perpendicular to the plane of the skull. As the needle touched the extraocular muscle cone, formed by the retractor bulbi muscle, a resistance and a slight ‘popping’ sensation was felt. In some cases, as the needle pressed against the cone the eye rotated dorsally and returned centrally after entering the cone.

#### 2.1.2. Group 2—Ultrasound-Guided Technique

RNBs were performed in seven cadaver heads (14 orbits) using ultrasound guidance (Figure 1) [10]. Ultrasonographic examination (US) was performed by a board-certified (ECVDI) veterinary radiologist (R.A.L.-E), while injection was performed by one of the two operators. This used a high frequency curvilinear transducer set in combination with an Epic 5 ultrasound machine (Philips C5-8 HD11 XE and Epic 5, Philips Medical Imaging and Healthcare, Eindhoven, The Netherlands). The US probe was positioned on the closed upper eyelid (transpalpebral US). The optic nerve was visualized in transverse, dorsal and sagittal planes. The needle was positioned using the dorsal (supraorbital) approach as mentioned above, in the supraorbital fossa, perpendicular to the skull. The needle was advanced using US visualization until it entered the cone formed by the retractor bulbi muscle.

#### 2.1.3. Group 3—Blind Technique

Following the US-guided nerve block, seven cadaver heads (14 cadaver orbits) were used for the blind technique again, as described in group 1, to evaluate the training effect of US guidance.

### 2.2. Computed Tomography

After the injections, the needles were kept in place and Computed Tomography (CT) imaging was performed using a helical, 64-slice sliding gantry CT scanner (SOMATOM Definition AS, Escel Edition, Siemens AG, Munich, Germany). The scanning protocol used was 140 kVp, 328 mAs and 512 × 512 matrix, 0.6 mm slice thickness, 0.5 s rotation time and 0.8 pitch. Bone (H60f) and soft-tissue (H31f) algorithm images were applied with reconstruction to, respectively, 1 mm and 2 mm slice thicknesses.

Image interpretation was performed blinded by a board-certified (ECVDI) veterinary radiologist (R.A.L.-E.), using a picture archiving and communication system (Impax, version 6.6.1.3304, N.V., Mortsel, Belgium). The CT findings were described as follows: needle positioning in the periorbita (centrally, peripherally, region of the globe/anteriorly, outside of periorbita/extraconal) and the distance of CM to the optic foramen (in mm). Central positioning was scored as optimal, peripheral positioning as suboptimal, while the region of the globe and extraconal were scored as undesired (Table 1).

### 2.3. Dissection

Following CT examination, dissection of all cadaver heads by an independent observer (J.C.M.S.) determined the injection site and dye distribution. The cornea was incised with a scalpel to evaluate whether methylene blue dye was visible intraocularly. At dissection, this inadvertent intraocular injection (yes or no) was recorded. Photographs of the dye distribution were taken for archiving purpose. 

### 2.4. Statistical Analysis

Statistical analysis was performed using open software R (3.6.3; R-Studio) and the package lme4 (version 1.1-21) was used for the linear mixed effect model and glm for logistic regression. A 95% confidence interval (C.I.) was reported and significance was set at *p* < 0.05. The mean distance of contrast distribution from the optic foramen and standard deviation (SD) in the different groups were calculated using excel (Microsoft Excel for Windows for Mac 2019, version 16.27, Microsoft, Albuquerque, NM, USA). As both operators were equally inexperienced, all data were pooled.

To compare the needle position between groups 1 and 2, groups 2 and 3 and groups 1 and 3, a logistic regression model was used. Needle positioning was classified as optimal if the needle was positioned centrally and undesired if the position was peripheral, in the region of the globe or outside of periorbita. The odds ratio (OR) was calculated from these models by calculating the exponential function of the model estimates.

A linear mixed model was used on the outcome variable contrast distribution distance from the optic foramen. Group (1, 2, 3) was used as a fixed effect and horse ID as a random effect. Normality of the residuals was checked using boxplots.

For the staining within the bulbus, a logistic regression model was used to compare groups 1, 2 and 3, and the OR was calculated from these models by calculating the exponential function of the model estimates.

## 3. Results

In total, 44 RNBs were performed. In group 1, 16 RNBs were placed; in group 2, 14; in group 3, 14 as well. Sample CT images (Figure 2 and Figure 3) show the needle position and the CM spreading.

Regarding the needle position in the blind technique, 37.5% of the injections were positioned in an optimal central position in the periorbita, 25.0% in a suboptimal peripheral position and 37.5% in the region of the globe (Table 2; Figure 4). No injections were placed outside the periorbita/extraconal. An optimal central position of the needle was seen in 75% of the US-guided samples in group 2, and a peripheral position in 25%. No injections were placed in the region of the globe or outside the periorbita (Table 2). In group 3, 71.4% of the injections were positioned centrally, 14.3% peripherally and 14.3% in the region of the globe, with no injections outside the periorbita (Table 2).

No significant difference existed regarding needle position between groups 1 and 2 (*p* = 0.056), groups 1 and 3 (*p* = 0.069), and groups 2 and 3 (*p* = 0.8).

The estimated marginal mean CM distribution distance from the optic foramen was 16.51 mm (C.I.: 6.60; 26.4) in group 1, 8.99 mm (C.I.: −1.59; 19.6) in group 2 and 10.91 mm (SD C.I.: 0.32; 21.5) in group 3. No significant difference existed in CM distribution distance between groups 1 and 2 (*p* = 0.17), between groups 2 and 3 (*p* = 0.30) or between groups 1 and 3 (*p* = 0.33).

The examples in Figure 5 show the dissection of an equine head, visualizing the injection site and dye distribution. Using the blind technique, in group 1, 18.75% (3/16) of the injections showed inadvertent intraocular administration, as staining within the bulbus was visible at dissection. Using the US-guided technique (group 2), no injections were intraocular. In group 3, the blind technique after US-guided training, one injection (7.1%, 1/14) was intraocular. Between groups 1 and 2, the difference was not significant (*p* = 0.99) with an OR of 0.0007.

## 4. Discussion

Locoregional techniques are gaining in their frequency of use as standing surgery in the equine patient becomes more and more common [10,18,19]. In most ophthalmic standing surgeries, the retrobulbar nerve block is a necessary local block, but it is not without risks [20]. A possible solution, the US-guided RNB, has been explored in horses [15], with only one study comparing the blind technique and the US-guided peribulbar technique (in inexperienced operators) [16]. In the current study, with inexperienced operators, there was no significant difference between the groups regarding needle position, but US-guided blocks showed a trend towards significantly improved needle position. This learning effect was mostly maintained during part 3 (group 3) of the study when the operators again performed the blind technique after being trained with US guidance during part 2 (group 2) of the study. Although our findings were not significant, we hypothesize that this is due to the relatively low number of cadaver heads used. A larger number of cadaver heads might have increased the power sufficiently to show significant effects of US guidance; however, this remains hypothetical and should be investigated in future studies.

We also found that when the needle is positioned in the region of the globe, there is a risk of inadvertent penetration of the globe and/or nerve injury or anterior spread of the local anesthetic. Anterior spread of the anesthetic may cause chemosis (swelling of the conjunctiva), an undesired effect of the RNB known in clinical practice [3,20,21]. In human as well as in equine practice, the prevalence of chemosis after RNB is unknown [20,22], although in a recent in vivo study of blind retrobulbar injections using eight horses, three quarters showed mild to moderate reversible chemosis for 2 to 24 h [21]. Chemosis does not seem to have a long-term effect, but it often interferes with surgical procedures and corneal ulceration can occur postoperatively [21]. The incidence of chemosis might even be under-reported in horses [21].

In our study, inexperienced operators using the blind technique caused intraocular injections in 18.75% of cadaveric eyes. In clinical equine practice this would be a devastating complication since globe preservation is often needed. In a comparable cadaveric study of RNBs in horses, no ocular perforation was described [15]. It also has not been reported in equine practice [20]. The differences between the existing literature and our study may be due to the inexperience of the persons performing the retrobulbar injection. In human medicine, the risk of ocular penetration is approximately between 1:1000 and 1:12,000 and is also related to the experience of the operator [3,4,23,24,25]. Thus, we advise using US-guided RNBs to reduce the risk of this complication, particularly when performed by inexperienced clinicians.

In our study, no evidence of penetration of the optic nerve was observed by US or CT. In a comparable ex vivo study by Morath et al., 2013, in one case (1/40) the optic nerve was punctured [15]. This might have been caused by the methodology used as they positioned the needle into the space enclosed by the retractor bulbi muscle, and in this position the needle is close to the optic nerve [15,20]. Penetration of the nerve/nerve sheath might lead to brain stem anesthesia or bilateral retrobulbar blocks, as described in humans, and is a dangerous complication [1,2,3]. When using the blind technique, this is a risk since needle position is difficult to assess. Fortunately, in equine clinical practice, the risk of inadvertent nerve penetration appears low, although few studies have evaluated this [20,26,27]. Again, these complications could be prevented, especially when operators are inexperienced, using US guidance.

Other complications of RNBs in horses that have been described in the literature include intra-meningeal injection, hematoma formation, optic neuritis, prolapse of the globe due to extraocular muscle blockade, orbital cellulitis and abscessation and orbital edema [11,20]. In our study, none of these complications were considered due to our ex vivo study design.

To avoid these complications, other techniques have also recently been described in horses, such as sub-Tenon’s injection and the US-guided peribulbar block [16,20,21,28].

In the sub-Tenon’s injection, a small conjunctival incision is made 5 mm from the limbus and a curved blunt sub-Tenon’s cannula is placed in a temporal-dorsal position following the equator of the globe and 7–10 mL of anesthetic is injected [28,29]. This is widely used in human patients and has also been described in dogs [29,30]. In horses (in an ex vivo design), this technique makes it possible to gain access to the sub-Tenon’s space and, as fluid distribution occurs in the posterior sub-Tenon’s space and intraconal, it might be a valid alternative to the RNB in horses [28]. Its disadvantages are the risk of chemosis and accidental globe puncture; experience with the injection technique seemed to be necessary to prevent the latter [28].

The US-guided peribulbar block was also recently described in horses [16]. In this technique, the needle is advanced in the supraorbital fossa by US guidance until the needle tip is adjacent to the extraocular muscle cone and 10 mL of anesthetic is injected. This US-guided peribulbar block is easy to perform without visible complications and all injections were described as being at the intended location [16]. The ‘in-plane’ US in this technique might provide better visualization of the needle tip and also, by placing them both in the supraorbital fossa, puts the needle and US out of vision of the horse. In the live animal this might prevent sudden movement [16]. This ‘in-plane’ US-guided approach might, thus, be an improvement over the US-guided RNB used in our study. Both techniques, the sub-Tenon’s injection and the peribulbar block, have the potential to be an alternative to retrobulbar block anesthesia in horses; however, their feasibility needs to be further evaluated in live patients.

No apparent effect of training with ultrasound was seen in our study as no significant difference was found between groups 1 and 3 and between groups 2 and 3. This might reflect the duration or intensity of the training or the low sample size of our study. As mentioned above, we did see a trend towards significantly improved needle position, showing a possible learning effect in inexperienced operators. We hypothesize that the relatively low number of cadaver heads used influenced our findings, with larger numbers possibly increasing the power and showing significant effects for US guidance. Additionally, as we had no intraocular injection using the US-guided technique, the use of US guidance seems promising.

In our study, US guidance was performed by a board-certified (ECVDI) veterinary radiologist (R.A.L.-E.), while the injection was performed by a second individual. In practice, the same clinician often performs both roles simultaneously. In either case, experience in both US guidance and injection under US guidance is essential to obtain the same results [6,7,8,31]. This is consistent with the study by Leigh et al., 2021, in which a US-guided peribulbar block was described and this US-guided technique was compared to the blind retrobulbar technique [16]. The RNB and US in that study were performed simultaneously and both operators had some experience using ultrasonography and moderate experience with the blind retrobulbar technique. Their results showed that, as all injections were placed correctly, the US-guided peribulbar block was easy to learn and to perform correctly. US-guided anesthesia might also be taught using an experimental model recently described in the literature [32]. In this way, inexperienced operators can improve the speed and accuracy of US-based needle placement [32].

The volume that was used for injection in our study (2.5 mL) was lower than the volume typically used in other studies [15,16,28]. This lower volume was selected to produce a more trustworthy result in dissection and because US guidance would decrease the volume of anesthetic injection solution required for an adequate nerve block [6]. It is important to minimize the injection volume in live horses, where high volumes might give a transient rise in intraocular pressure or an increased risk of vagal stimulation [3,21,33,34].

The ideal injection volume to achieve adequate spread in horses is unknown, although the recommended injection volume for RNBs ranges from 10 to 12 mL [9]. In the study by Morath et al., 2013, different injection volumes (4, 8 and 12 mL) were compared [15]. The injection volume of 4 mL had a statistically significant effect on outcome, with no difference between 8 and 12 mL [15]. On the basis of these results, an injection volume of 8 mL was advised [15]. No direct comparison to our study can be made as the study by Morath et al., used only Warmblood horses and we used only Shetland ponies. In humans, the recommended injection volume is adjusted to the patient’s orbital dimensions [35].

Our study, with a relatively small sample size, used contrast medium instead of local anesthetic solution and used only cadaver heads instead of live animals. However, despite the relatively small sample size, the observational results suggest that US guidance decreases the risks of RNB placement and shows the importance of US guidance. These results also emphasize the effect of training by US guidance for inexperienced operators. Additionally, as mentioned above, if we had had access to a larger number of cadaver heads, our results might have increased the power and shown significant effects of US guidance.

Since the contrast medium used in our study is more viscous than local anesthetic solution, the distribution of local anesthetic might be better in live animals than in our study.

Using only cadaver heads also has the disadvantage that movement and normal tissue distribution, as happens in the live horse, is not present. At the same time, in live horses, adequate sedation or general anesthesia minimizes the risk of movement causing problems with visualization of the optical cone by US or inadvertent injection into important structures [20]. In our study, both fresh and thawed cadaver heads were used and all frozen cadaver heads were thawed thoroughly. Using only fresh cadaver heads may influence the results as US guidance might have been more accurate in fresh cadaver heads. However, as all groups contained both (fresh as well as thawed cadaver heads), we do not expect the results to be different.

Finally, one could argue that, in our study, time could have influenced the outcome as the inexperienced operators would naturally become more experienced over time. As between injections in groups 1 and 2 around six weeks had passed and between groups 2 and 3 around three months had passed, and the number of injections in each group was limited, we believe that this effect was limited.

However, further studies in live horses are warranted to assess the safety and efficacy of the US-guided technique and to compare this technique to the blind technique.

## 5. Conclusions

Our study demonstrates that US guidance may successfully increase accuracy in studies using higher numbers of skulls and decrease the rate of complications associated with retrobulbar nerve blocks, even when performed by inexperienced operators. However, to better quantify this possible effect, more studies are warranted. Furthermore, for a more general clinical recommendation, further validation in live horses is necessary.

## Figures and Tables

**Figure 1 animals-12-00154-f001:**
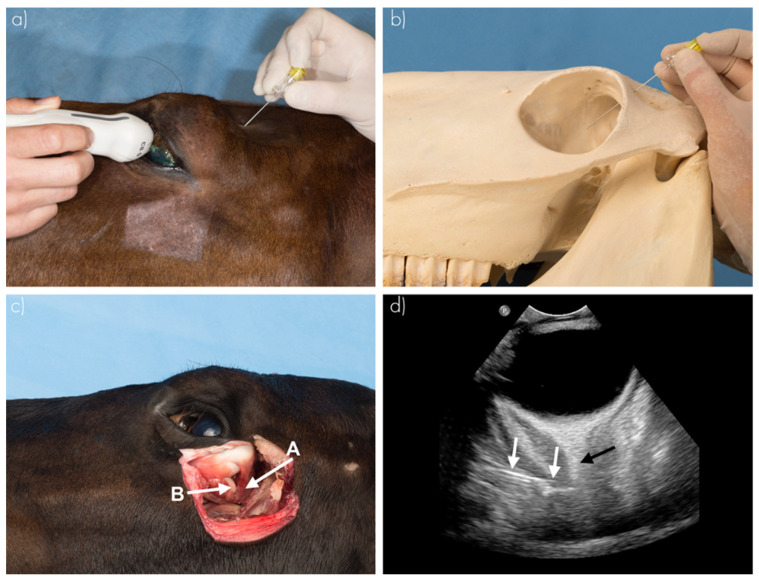
Ultrasound-guided retrobulbar nerve block (RNB). (**a**) Positioning of the ultrasound probe and the needle on the head. (**b**) Needle placement with respect to the bony landmarks. (**c**) Extrinsic straight eye muscles (A) within the covering fascia (cone) and the optic nerve in the center of the straight muscles (B). (**d**) Ultrasound image of retrobulbar needle placement and associated structures (white arrows show needle placement with tip at the height of the right arrow, black arrow shows the optic nerve) [10].

**Figure 2 animals-12-00154-f002:**
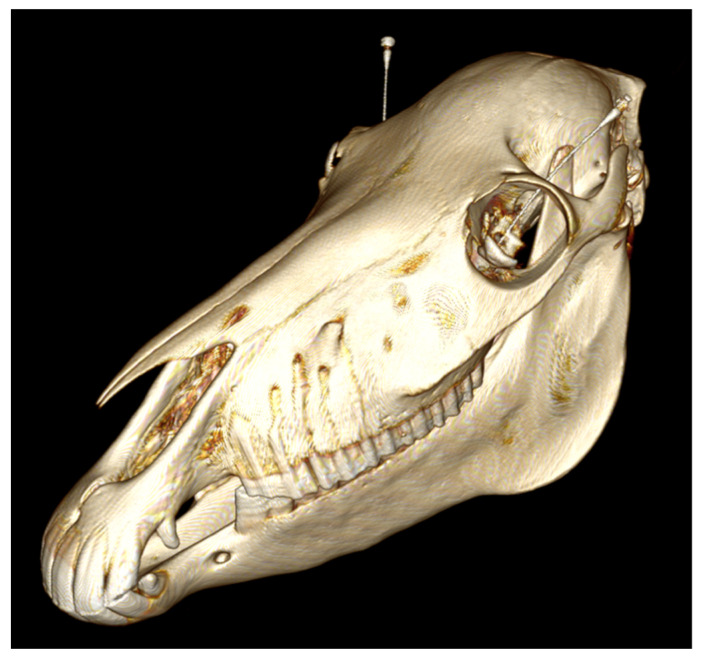
3D reconstruction of a CT image of a cadaver head after the RNB with the needles in place. Bilaterally, the spinal needles are visible in the optical cone.

**Figure 3 animals-12-00154-f003:**
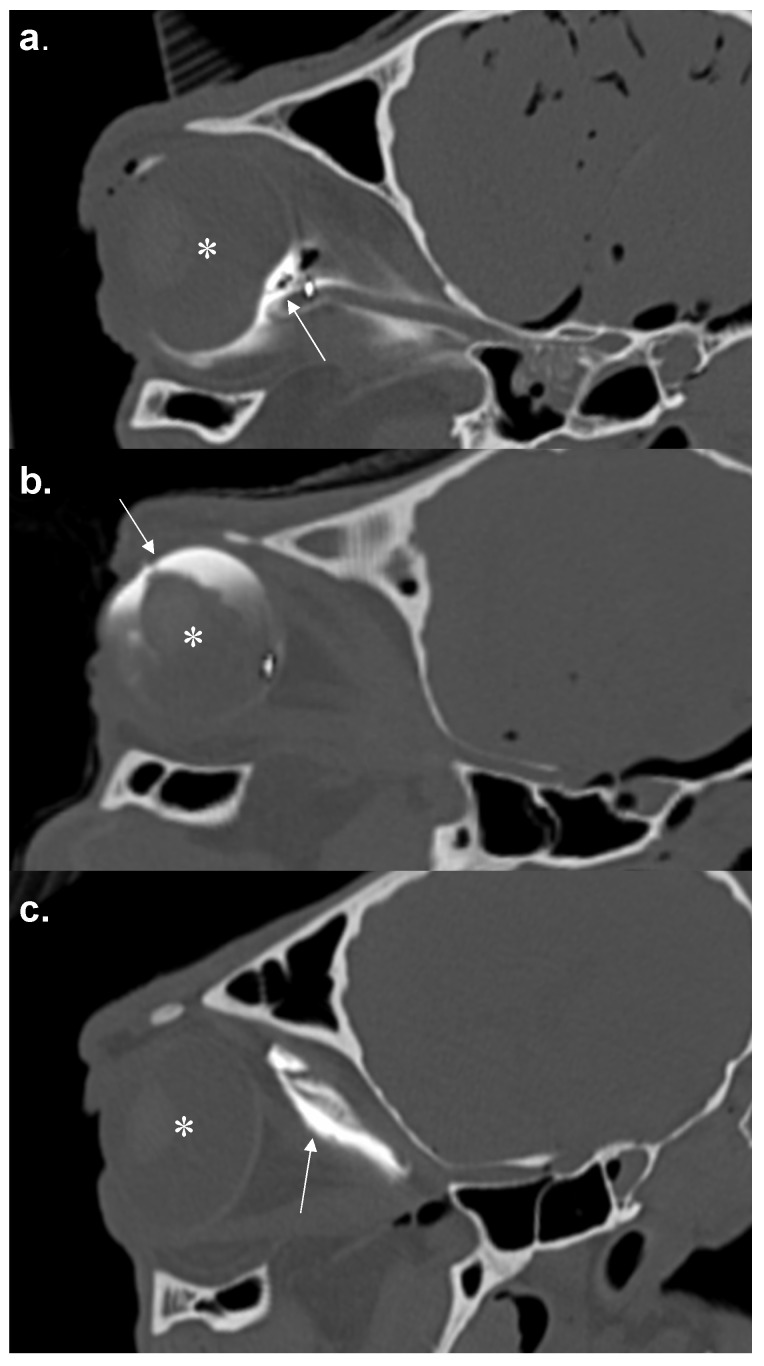
Oblique parasagittal (aligned along the optic cone) reconstructions of the computed tomographic (CT) scans of three optic cones with central (**a**), anterior (**b**) and peripheral (**c**) distribution of contrast medium. The arrows indicate the distribution of contrast medium (CM) and the asterisk the eyeball.

**Figure 4 animals-12-00154-f004:**
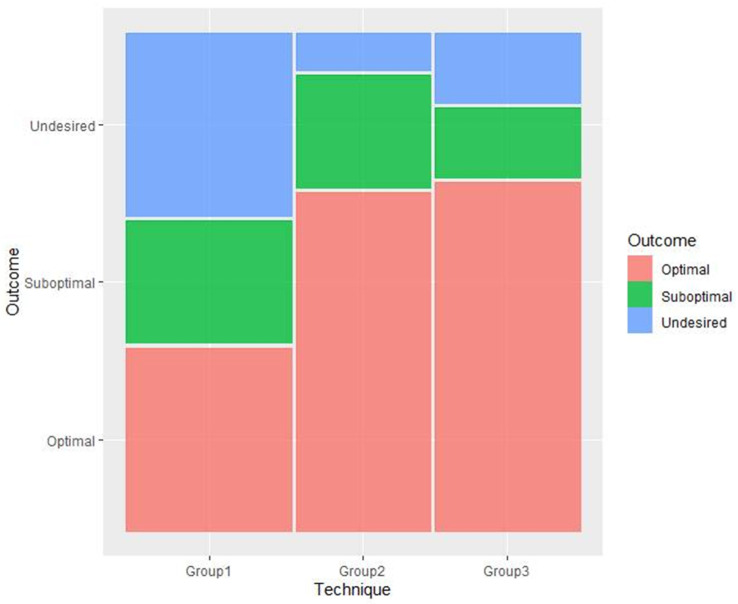
Schematic overview of undesired (region of the globe/anterior; blue), suboptimal (peripherally; green), and optimal (central) position of the needle (red) in the different groups (group 1–3).

**Figure 5 animals-12-00154-f005:**
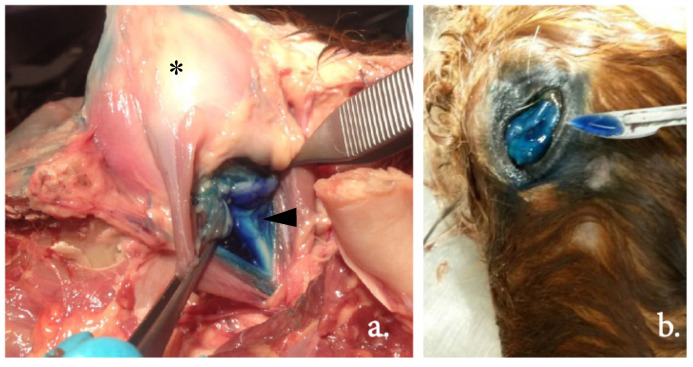
Methylene blue dye distribution at dissection of one of the equine cadaver heads in blue: (**a**) The dye was distributed in the periorbita (arrow head) with, centrally, the optic nerve visible; the asterisk indicates the eyeball.; (**b**) An example of intraocular dye in the eyeball.

**Table 1 animals-12-00154-t001:** Scoring system for needle positioning at Computed Tomography (CT).

Position	Category	Criteria
Central	Optimal	Needle positioned centrally within the periorbita
Peripheral	Suboptimal	Needle positioned peripherally within the periorbita
Region of globe	Undesired	Needle positioned anteriorly within the region of the globe
Extraconal	Undesired	Needle positioned outside of cone of retrobulbar muscle

**Table 2 animals-12-00154-t002:** Variables measured in the different groups (1–3): The number of RNBs that showed staining of methylene blue dye within the bulbus at dissection (positive) versus the number of injections that did not show staining at dissection (negative); the position of the needle within the periorbita measured by CT; contrast distribution of the CM in mm from the optical foramen measured by CT.

	Staining within Bulbus		Needle Positioning (*n*)				Contrast Distribution	Total (*n*)
	Positive	Negative	Optimal	Suboptimal	Undesired		Mean mm from Optical Foramen ± SD	
			Central	Peripheral	Anterior	Extraconal		
**Group 1** **blind**	3 (18.75%)	13 (81.25%)	6 (37.5%)	4 (25%)	6 (37.5%)	0 (0%)	16.51 ± 4.21	16
**Group 2** **US**	0 (0%)	14 (100%)	9 (75%)	3 (25%)	0 (0%)	0 (0%)	8.99 ± 1.12	14
**Group 3** **blind**	1 (7.14%)	13 (92.86%)	10 (71.4%)	2 (14.3%)	2 (14.3%)	0 (0%)	10.91 ± 3.08	14

## Data Availability

The data that support the findings of this study are openly available in DataverseNL at www.dataverse.nl, https://doi.org/10.34894/LDUI10.

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
