# Peer review of "Comparing Blind and Ultrasound-Guided Retrobulbar Nerve Blocks in Equine Cadavers: The Training Effect"

_animals, 2022, doi:10.3390/ani12020154_

Round 1

Reviewer 1 Report

Dear authors,

thanks for your manuscript. It is interesting, but there are some major points that must to be addressed.

In the summary (line 19) you talk about anterior spread. Please, be more precise. Do you mean the anterior chamber of the eye? The globe itself? etc.

Line 29. I strongly disagree with your statement of a 'trend' existing between groups 1 and 2, and 1 and 3. All the results in your study are non-significant. We must be honest with the readers and with ourselves. Then, after stating that, you can comment on tendencies, etc (see comments later).

Line 40. You must give the % with the 95% confidence interval.

Line 46. Change could for might

Line 80. How many heads were used fresh and how many thawed. Was any statistical analysis done to try to determine if there is a difference or limitation in the fact that 1 group was done with fresh heads and the rest with thawed? Maybe add it as a limitation of the study.

Line 83. I would say somewhere here that the retrobulbar' injections' will be referred in the rest of the manuscript as retrobulbar blocks. In fairness, they are not blocks because you did not use local anaesthetic and you performed them in cadavers.   

Line 84. Why did you use 2 operators? when you did nothing with that information later on? At the very least, you also have to let us know if the retrobulbar blocks were divided equally between the operators or not. Currently we don't know.

Line 91. Was the total dose 2.5 mL? this is not clear til the discussion.

Line 98. Here is not clear if the US was done by the DipECVDI and the operators performed the needling. You don't make it clear til the discussion. Please, add something in here.

Line 135. When you mention that photographs of the dye were taken. Please, tell us why to make it more relevant. As it sounds now, it is irrelevant information. Was it for further analysis? or for archiving them?

Line 137. Why a sample size calculation was not performed before starting the study (or, truly, at any point)? Why the inter operators results were not compared before compiling them together? We don't know if one of the operators was better than the other or one improved more than the other. You should have compared them between them, and if not significantly different between them, then join the results together for further analysis.

Line 194. Give the 95% confidence interval in all the results.

Line 203. In fairness, none of the groups are significant. You should start there before saying that there is a tendency or trend. You also insist in saying that more animals would confirm that trend, but this is far from being true. You cannot know that. Why was not there a sample size calculated? In fact, I performed a quick sample size calculation with your results and shows that you would need only 27 blocks to detect a difference between group 1 and 2. So, adding more animal would not increase your chances for that p value to become significant.

Line 273. Add THE before literature.

Line 312. That information should be in the materials and methods section.

Line 355. All this paragraph is full of results. If you want to discuss the time frame between groups, you must provide this information beforehand in the results section.

Overall, the research is interesting, but there are few relevant gaps and questions that must be answered before being published.

Author Response

Dear Reviewer, 

Thank you for the critical comments that have helped to strengthen the manuscript. 

We have made revisions to the manuscript according to the recommendations and have given the specific details about the changes to each of the comments below. We provided our point-to-point response in blue, italic font. If we did not make the recommended change we have also provided justification as to why.  

Please see the attached word file for the response. 

Reviewer 2 Report

GENERAL COMMENTS: in my opinion the manuscript should be significantly integrated. Materials and methods are reductive and not very explanatory. Results should be expressed with greater clarity and, furthermore, all figures should be integrated with specific legends describing what is represented. Discussions are rich in content, but there is often no correlation or continuity between one paragraph and the next. In general, the manuscript could be improved.

INTRODUCTION

Line 52:  change “in the standing horse” with “during standing procedures in horses”

Line 57: change “they have become more common” with “currently, they are widely used to improve….” (improve the sentence)

MATERIAL AND METHODS

Line 78: delete “humanely”

Line 80: please specify how many heads were fresh and how many were frozen. Don't you think this difference can affect the result? (especially in the group which you used the ultrasonographic method)

Line 92-93: how much time has passed from the block to the cadaverous evaluation? Was it the same in all three groups? specify the elapsed time and any differences between groups.

Line 95-96: please describe the technique used specifying in detail the times and the manual skills performed.

Line 99-100: why was the ultrasound visualization done by a specialist? don't you believe that anatomical visualization is part of the execution capability of the block?

Line 98-102: see line 95-96

Line 133-135: please describe in detail the dissection technique

RESULTS

Line 156: did you perform a statistical calculation to obtain the sample power? we recommend that you do it.

DISCUSSION

Line 230-231: change “in most standing surgeries” with “in most ophthalmic standing surgeries”

Line 236: certainly, in group 3 there was an improvement in the operator skills but, in the discussions, I believe that you should specify that it is always advisable to rely on the safest technique, with a reduced risk of complications for our patient.

Line 258-259: the references are very old (1991-2006). you should enter more recent data.

Line 303-304

FIGURE: Please insert in each figure a legend and a corresponding description of the structures displayed, indicating them with arrows.

Author Response

Dear Reviewer, 

Thank you for the critical comments that have helped to strengthen the manuscript. We have made revisions to the manuscript according to the recommendations and have given the specific details about the changes to each of the comments below. We provided our point-to-point response in blue, italic font. If we did not make the recommended change we have also provided justification as to why.

Please see the attachment with our response. 

Round 2

Reviewer 1 Report

Thank you for addressing the previous comments.

I don't see anything else to bring up.

All the best.

Author Response

Dear Reviewer,

Many thanks for your positive feedback and for your acceptance of the article. 

Reviewer 2 Report

I believe that the structure of the text could still be improved (many words and phrases were repetitive, so please correct them) and the paragraphs could be linked more fluidly. However, in general, the clarity of the concepts has been considerably improved and the description of the techniques have been significantly increased.

Simple Summary

Line 24-25: change with “Our study aimed to compare the blind and ultrasound guided approaches to retrobulbar block placement in horses and to evaluate success and complication rates, analysing the effect of training of ultrasound guidance”

Introduction

Line 56: please insert the bibliography in horses at the end of the paragraph

Line 69: please change with “if ultrasound guidance is significantly useful in RNB placement”

Material and Methods

Line 73: please correct the double parenthesis error and write more clearly

Line 75: “All ponies were clinically healthy” please specify that they did not have any eye diseases that may alter the results of the study

Line 74-75: to not be repetitive I would change the sentence “were euthanized for reasons other than this study. All ponies were clinically healthy. All ponies were euthanized and heads were removed from the trunk following euthanasia” with “were euthanized for reasons other than this study and heads were removed from the trunk”.

Line 84: change “Each performed the RNB on one eye on each cadaveric head, with laterality randomized” with “Each performed RNB on one eye of each head, randomly choosing the left or right side”

Line 87: to not be repetitive I would change “all injections” with “the injections”

Line 100: to not be repetitive (you repeat several times "needle")  I would change  “the dorsal (supraorbital) approach [17]. The spinal needle was placed caudal” with “the dorsal (supraorbital) approach [17] and it was placed caudal…”

Statistical analysis

What statistic did you use to analyse the yes / no variables?

Discussion

Line 241: please insert references

Author Response

Dear Reviewer,

Thank you again for the critical comments that have helped to strengthen the manuscript. We have made revisions to the manuscript according to the recommendations and have given the specific details about the changes to each of the comments below. We provided our point-to-point response in blue, italic font. If we did not make the recommended change we have also provided justification as to why.

Comments and Suggestions for Authors

I believe that the structure of the text could still be improved (many words and phrases were repetitive, so please correct them) and the paragraphs could be linked more fluidly. However, in general, the clarity of the concepts has been considerably improved and the description of the techniques have been significantly increased.

We would like to thank the reviewer for the encouraging comment and the constructive feedback. We have attempted to fully answer the concerns below.

Our English academic editor Linda McPhee (Lindamcpheeconsulting.com) helped us again to improve the structure of the text and we removed repetition. We hope the reviewer agrees with the changes and structure of the manuscript now. 

We have made revisions to the manuscript according to the recommendations and have given the specific details about the changes to each of the comments below. We provided our point-to-point response in blue, italic font. If we did not make the recommended change we have also provided justification as to why.  

Simple Summary

Line 24-25: change with “Our study aimed to compare the blind and ultrasound guided approaches to retrobulbar block placement in horses and to evaluate success and complication rates, analysing the effect of training of ultrasound guidance”

Edits made as suggested by the reviewer (Simple Summary, line 24-26, page 1).

Introduction

Line 56: please insert the bibliography in horses at the end of the paragraph

We have inserted the bibliography of horses at the end of the paragraph as suggested by the reviewer (Introduction, line 62-64, page 2).

Line 69: please change with “if ultrasound guidance is significantly useful in RNB placement”

We have changed this sentence into: “The aim of our study was thus to compare the blind technique with the US-guided technique in horses and to determine whether ultrasound guidance is significantly useful in RNB placement.” (Introduction, line 66-68, page 2).

Material and Methods

Line 73: please correct the double parenthesis error and write more clearly

We have changed the sentence to: “Twenty-two clinically healthy Shetland ponies, comprising 21 mares and 1 gelding, with a mean age of 6.4 years (range 4-10.5 years) and a mean body weight of 167.4 kg (range 138-204 kg), without signs of ophthalmic disease, were euthanized for reasons other than this study and afterwards heads were removed from the trunk.” (Material and Methods, Animals, line 72-75, page 2).

Line 75: “All ponies were clinically healthy” please specify that they did not have any eye diseases that may alter the results of the study

Thank you for this comment. We have changed the sentence to: “Twenty-two clinically healthy Shetland ponies, comprising 21 mares and 1 gelding, with a mean age of 6.4 years (range 4-10.5 years) and a mean body weight of 167.4 kg (range 138-204 kg), without signs of ophthalmic disease, were euthanized for reasons other than this study and afterwards heads were removed from the trunk.” (Material and Methods, Animals, line 72-75, page 2).

Line 74-75: to not be repetitive I would change the sentence “were euthanized for reasons other than this study. All ponies were clinically healthy. All ponies were euthanized and heads were removed from the trunk following euthanasia” with “were euthanized for reasons other than this study and heads were removed from the trunk”.

See the sentence in the previous comments (Material and Methods, Animals, line 72-75, page 2).

Line 84: change “Each performed the RNB on one eye on each cadaveric head, with laterality randomized” with “Each performed RNB on one eye of each head, randomly choosing the left or right side”

Edits made as suggested by the reviewer (Material and Methods, Animals, line 83-84, page 2).

Line 87: to not be repetitive I would change “all injections” with “the injections”

Edits made as suggested by the reviewer (Material and Methods, Animals, line 86, page 2).

Line 100: to not be repetitive (you repeat several times "needle")  I would change  “the dorsal (supraorbital) approach [17]. The spinal needle was placed caudal” with “the dorsal (supraorbital) approach [17] and it was placed caudal…”

The sentence was changed into: “RNBs were performed using the blind technique in eight cadaver heads (16 cadaver orbits). The spinal needle was placed using the dorsal (supraorbital) approach and placed caudal to the center of the dorsal orbital rim [17]. It was advanced into the retrobulbar space perpendicular to the plane of the skull.” (Material and Methods, Group 1 – Blind technique, line 96-99, page 3).

Statistical analysis

What statistic did you use to analyse the yes / no variables?

For the staining within the bulbus (yes / no variable), a logistic regression model was used to compare groups 1, 2 and 3 and the OR was calculated from these models by calculating the exponential function of the model estimates.

Discussion

Line 241: please insert references

We have added the following references to the sentence as suggested by the reviewer (Discussion, line 241, page 8):

  1. Kumar, C.M.; Dowd, T.C. Complications of ophthalmic regional blocks: Their treatment and prevention. Ophthalmologica 2006. 220, 73–82. doi: 10.1159/000090570

  1. McKinney, R.A. The retrobulbar block: A review of techniques used and reported complications. Equine vet. Educ. 2020. 33, 332-336. doi: 10.1111/eve.13351

  1. Yang, V.Y.; Eaton, J.S.; Harmelink, K.; Hetzel, S.J.; Sanchez, A.; Lund, J.R; Smith, L.J. Retrobulbar lidocaine injection via the supraorbital fossa is safe in adult horses but produces regionally variable periocular anesthesia. Equine Vet J. 2021 Epub ahead of print. doi: 10.1111/evj.13496